# Understanding Chemotherapy-Induced Thrombocytopenia: Implications for Gastrointestinal Cancer Treatment

**DOI:** 10.3390/curroncol32080455

**Published:** 2025-08-14

**Authors:** Supriya Peshin, Adit Dharia, Ehab Takrori, Jasmeet Kaur, Kannan Thanikachalam, Renuka Iyer

**Affiliations:** 1Norton Community Hospital, Norton, VA 24273, USA; supriyapeshin720@gmail.com; 2HCA Florida Healthcare, USF Morsani College of Medicine GME Oak Hill Program, Orlando, FL 32801, USA; aditdha@gmail.com; 3College of Medicine, Alfaisal University, Riyadh 11533, Saudi Arabia; etakrori@gmail.com; 4Fox Chase Cancer Comprehensive Center, Philadelphia, PA 19111, USA; jasmeet.kaur@tuhs.temple.edu; 5Roswell Park Comprehensive Cancer Center, Buffalo, NY 14203, USA; kannan.thanikachalam@roswellpark.org

**Keywords:** chemotherapy-induced thrombocytopenia, gastrointestinal cancers, FOLFOX, FOLFIRINOX, oxaliplatin, gemcitabine, platelet transfusion, thrombopoietin receptor agonists, romiplostim, eltrombopag

## Abstract

Many patients with gastrointestinal cancers experience a drop in their platelet count due to chemotherapy, a condition known as chemotherapy-induced thrombocytopenia. This can increase the risk of bleeding and may force doctors to delay or reduce cancer treatment, which can affect how well it works. Unfortunately, this condition is often overlooked, and there are no clear guidelines on how best to manage it. In this article, we explain what causes this problem, how often it occurs, and what treatments are currently available. We also highlight new drugs being studied and areas where more research is needed. Our goal is to raise awareness and encourage further investigation so that doctors can better protect patients and ensure they receive the full benefit of their cancer treatments.

## 1. Introduction

Cancer continues to be a major global health challenge, with an estimated 19.3 million new cases and 1 million cancer-related deaths reported annually [1]. Despite substantial advancements in cancer therapy, including immunotherapy and molecular-targeted treatments, systemic chemotherapy remains the cornerstone of treatment for gastrointestinal malignancies. Among the standard chemotherapy regimens for gastrointestinal cancers, oxaliplatin-based protocols, such as S-1 plus oxaliplatin, oxaliplatin with capecitabine, and oxaliplatin combined with leucovorin and fluorouracil (FOLFOX), continue to be widely utilized. These regimens are frequently employed as first- and second-line therapies for gastrointestinal tract tumors, making their use nearly unavoidable. In pancreatic cancer, the FOLFIRINOX regimen, comprising oxaliplatin, irinotecan, leucovorin, and 5-fluorouracil, has emerged as an intensive first-line option, particularly for patients with good performance status, offering improved survival at the cost of increased toxicity [2]. Unlike prior broad oncology reviews, this study synthesizes recent randomized and real-world data on thrombopoietin receptor agonists (TPO-RA), most notably the phase III avatrombopag study and global RECITE trial of romiplostim presented at the ASCO 2025, to aid in presenting an algorithm tailored specifically to gastrointestinal malignancies, which happens to be one of the largest solid tumor subgroups affected by chemotherapy-induced thrombocytopenia (CIT) [3,4].

A major complication of chemotherapy is CIT, which results from the suppression of bone marrow megakaryocytes, leading to a decline in peripheral platelet counts below 100 × 10^9^/L [5]. In a large U.S. retrospective cohort of 15,521 solid-tumor patients, the three-month incidence of CIT was 13–15%, while severe CIT as per the National Cancer Institute’s Common Terminology Criteria for Adverse Events (CTCAE) was 4% in grade 3 and 2% in grade 4 patients [6]. Among patients with solid tumors, CIT is most commonly observed in non-small cell lung cancer (25%), ovarian cancer (24%), and colorectal cancer (18%) [7]. A retrospective hospital-based study conducted in the Netherlands, which included over 600 adult patients receiving chemotherapy for solid tumors, reported that 22% of patients developed thrombocytopenia (platelet count < 100 × 10^9^/L) [8]. The highest incidence was observed in patients receiving carboplatin monotherapy (82%) and oxaliplatin monotherapy (50%). Additionally, combination regimens involving carboplatin (58%), gemcitabine (64%), and paclitaxel (59%) were also associated with an increased risk of thrombocytopenia [8].

CIT remains a frequent and clinically significant complication in cancer patients undergoing cytotoxic chemotherapy, affecting approximately 20–25% of individuals with solid tumors [7,9]. CIT’s incidence, severity, and duration vary widely based on the specific chemotherapeutic agents and their administered doses [7,9]. Notably, regimens containing gemcitabine and platinum-based compounds are associated with a particularly high risk of thrombocytopenia, defined as platelet counts dropping below 150 G/L. Reported thrombocytopenia rates in patients receiving gemcitabine-based therapy are as high as 64%, while those undergoing treatment with platinum-containing agents experience rates of 56% [7].

## 2. Pathophysiology

Outlined below are the distinct and interconnected pathophysiological processes that contribute to the development of CIT, reflecting its complex biological underpinnings (Figure 1).

### 2.1. Direct Cytotoxicity on Megakaryocytes and Progenitor Cells

Chemotherapeutic agents, particularly alkylating agents (e.g., cyclophosphamide), platinum-based drugs (e.g., oxaliplatin, cisplatin), and antimetabolites (e.g., 5-fluorouracil, methotrexate), are known to damage actively proliferating cells. Since HSPCs and megakaryocyte progenitors in the bone marrow undergo rapid division, they are particularly susceptible to chemotherapy-induced apoptosis and DNA damage, leading to platelet-producing cell depletion and thrombocytopenia [10].

A notable example is oxaliplatin-induced thrombocytopenia, which can occur acutely due to immune-mediated destruction or chronic marrow suppression. Oxaliplatin-based chemotherapy regimens, commonly used for gastrointestinal cancers, have been associated with acute thrombocytopenia, sometimes leading to treatment delays or dose reductions [11] (Table 1).

### 2.2. Disruption of the Bone Marrow Microenvironment

The bone marrow niche supports hematopoietic stem cell (HSC) maintenance and differentiation and consists of stromal cells, endothelial cells, and cytokines that regulate hematopoiesis. Chemotherapy alters this microenvironment by the following mechanisms:Depleting supportive stromal cells, impairing their ability to secrete essential growth factors (e.g., TPO, interleukins).Inducing oxidative stress and inflammation, disrupting normal HSC function.Damaging endothelial cells, which are crucial for sustaining the BM niche and protecting stem cells from toxic insults [10].

As a result, megakaryopoiesis is severely compromised, leading to inadequate platelet production and a higher risk of bleeding complications. Furthermore, studies have shown that platinum-based chemotherapy and targeted therapies can cause long-term suppression of bone marrow progenitor cells, further worsening CIT in cancer patients [23] (Table 2).

### 2.3. TPO Dysregulation and Impaired Megakaryopoiesis

TPO is crucial for platelet production, as it stimulates the differentiation and maturation of megakaryocytes. Under normal conditions, when platelet levels drop, TPO levels rise to compensate. However, chemotherapy-induced myelosuppression disrupts this process in several ways:TPO levels may be insufficiently elevated to compensate for platelet loss.Megakaryocyte precursors may have reduced responsiveness to TPO signaling due to drug-induced cellular damage.The liver, which produces TPO, may also be affected by chemotherapy, leading to suboptimal TPO production [24].

These factors collectively exacerbate thrombocytopenia and delay platelet recovery after chemotherapy. Studies have explored the use of TPO receptor agonists (e.g., eltrombopag, romiplostim) to mitigate CIT, with promising results in clinical trials [25] (Table 3).

### 2.4. Immune-Mediated Platelet Destruction

Certain systemic therapies, particularly immune checkpoint inhibitors and targeted therapies, can inadvertently trigger an immune response against platelets or megakaryocytes, leading to their accelerated clearance from circulation. This is observed in some patients treated with immune checkpoint inhibitors (e.g., pembrolizumab, nivolumab), which can cause immune thrombocytopenia (ITP)-like syndromes in addition to myelosuppressive effects.

Furthermore, drug-associated thrombocytopenia has been linked to platelet-reactive antibodies, which accelerate platelet destruction in the spleen. Studies suggest that drug-induced thrombocytopenia can occur through immune-mediated pathways, particularly with certain chemotherapy drugs and monoclonal antibodies [26] (Table 4).

### 2.5. Gemcitabine-Induced Thrombotic Microangiopathies (TMA)/Atypical HUS

Gemcitabine-induced thrombotic microangiopathy (GiTMA) is a rare but potentially life-threatening complication observed in patients receiving gemcitabine-based chemotherapy, particularly for pancreatic and other gastrointestinal cancers. The incidence of GiTMA is estimated to range between 0.015% and 0.31% of treated patients, though some single-center studies suggest higher rates in heavily pretreated populations [27]. The pathogenesis is thought to involve direct endothelial injury, resulting in platelet activation, microvascular thrombosis, and complement activation particularly through the alternative pathway, resembling atypical hemolytic uremic syndrome (aHUS) [28,29]. Clinically, patients present with the triad of microangiopathic hemolytic anemia, thrombocytopenia, and acute kidney injury, often accompanied by hypertension and proteinuria. Management includes prompt discontinuation of gemcitabine and supportive care, such as dialysis when indicated. While the role of plasma exchange remains unclear, eculizumab, a complement C5 inhibitor, has shown promise in selected patients, particularly those with complement dysregulation [30].

## 3. Incidence of CIT-Induced Thrombocytopenia

According to the NCCN guidelines, CIT is defined as a platelet count of <100 × 10^9^/L persisting for ≥3 to 4 weeks after the last chemotherapy administration and/or causing delays in chemotherapy initiation due to thrombocytopenia. A threshold of <100 × 10^9^/L is clinically significant, as it correlates with higher risks of treatment modifications, including dose delays, reductions, or discontinuations, and an increased likelihood of recurrence in future chemotherapy cycles [31].

The Common Terminology Criteria for Adverse Events (CTCAE) categorizes thrombocytopenia into four grades based on platelet counts: Grade 1: 75–100 × 10^9^/L, Grade 2: 50–75 × 10^9^/L, Grade 3: 25–50 × 10^9^/L, and Grade 4: <25 × 10^9^/L. The clinical impact of thrombocytopenia aligns with these grades, ranging from mild, asymptomatic cases with petechiae to life-threatening hemorrhagic events, such as intracranial or gastrointestinal bleeding [32,33]. The grades are also clinically categorized based on bleeding severity. Grade 1 corresponds to petechiae, Grade 2 to mild blood loss, Grade 3 to gross blood loss, and Grade 4 to debilitating blood loss [34] (Table 5).

Prospective data show substantial heterogeneity in CIT risk across standard GI cancer regimens. In a pooled analysis of two phase III trials studying 2145 patients with colorectal cancer, it was evident that the incidence of CIT occurred in 37% of FOLFOX4 (fluorouracil, leucovorin, oxaliplatin) cycles versus 4% of FOLFIRI (fluorouracil, leucovorin, irinotecan) cycles. Grade 3/4 events were likewise higher with FOLFOX4 (2%) than with FOLFIRI (<1%). CIT events that led to chemotherapy delay, dose modification, or discontinuation were 62% in FOLFOX4 versus 32% in FOLFIRI [35].

A multicenter Chinese series of 58 GI cancer patients who developed CIT provided further context. When treated with oral avatrombopag (40–60 mg daily for five days), 53.4% achieved a platelet count > 100 × 10^9^ or doubled their baseline within 5 days, 82.8% by day 11, and 96.6% by day 18, without drug-related toxicity [36]. The high rate of rescue pharmacotherapy, coupled with rapid platelet recovery observed, underscores both the frequency of clinically significant CIT in GI malignancies and the unmet need for proactive, regimen-tailored mitigation strategies.

Although platelet count is a key predictor of bleeding risk, other factors significantly contribute to bleeding complications in CIT. Platelet function, the rate of platelet decline, infections, kidney insufficiency, underlying coagulopathies, and the concomitant use of antithrombotic agents all modulate the overall bleeding risk [18]. These factors may explain why some patients with moderate thrombocytopenia experience severe bleeding while others with very low platelet counts remain asymptomatic. Further research is needed to refine risk assessment models beyond platelet thresholds and incorporate functional platelet activity and patient-specific risk factors into CIT management strategies.

## 4. Discussion

Among solid tumors, gastrointestinal malignancies are particularly prone to CIT, due to both disease-related factors and the intensive chemotherapy regimens used for treatment, contributing to treatment delays, dose modifications, and increased bleeding risk. The incidence of CIT varies depending on tumor type, patient characteristics, and treatment regimen, with some studies reporting rates ranging between 13–40% in patients receiving chemotherapy for GI cancers [18,32]. The risk of thrombocytopenia is further exacerbated in patients with underlying liver dysfunction, which is common in GI cancers such as hepatocellular carcinoma and metastatic colorectal cancer. Additionally, thrombocytopenia in these patients is not only a consequence of myelosuppressive chemotherapy but may also be influenced by splenic sequestration and compromised thrombopoiesis due to liver disease [37].

Given the high incidence of CIT in GI cancers, it is essential to evaluate which chemotherapy regimens contribute most significantly to this complication and how they impact treatment outcomes. Fluoropyrimidines (5-fluorouracil and capecitabine), gemcitabine, platinum-based agents (oxaliplatin and cisplatin), and irinotecan are among the most frequently used cytotoxic drugs in the treatment of GI malignancies, either as monotherapy or in combination regimens.

Multiple retrospective studies have investigated CIT in solid tumors. In a large retrospective study by Shaw et al., 13% of patients with solid tumors developed thrombocytopenia, with the highest incidence observed in patients receiving gemcitabine and platinum-based therapies [32]. Similar findings were reported in a retrospective study of 47,159 patients, reinforcing the association between CIT and both platinum-based and gemcitabine-containing regimens [7].

Five high-profile reviews published since 2022 have shaped current thinking of CIT. Kuter et al. reviewed studies across all solid tumors and confirmed platinum-, gemcitabine-, and temozolomide-based regimens drive the highest CIT rates; they highlight that in a multicenter romiplostim series, 71% of patients achieved platelet recovery and 79% avoided chemotherapy delays and dose reductions [18]. Gao et al. provided a mechanistic literature review, citing highly variable CIT incidence in 10–68% of patients with solid tumors, indicating that robust TPO-RA randomized data were still lacking at the time of publication [6]. Soff et al. issued International Society on Thrombosis and Hemostasis (ISTH) guidance based on randomized and observational cohorts focusing on romiplostim. Weekly dosing corrected platelet counts in 93% of patients versus 12.5% of controls within 3 weeks, and the accompanying multicenter study documented a 79% reduction in dose delays and an 89% elimination of transfusions when weekly schedules were used [31]. Al-Samkari translated this efficacy signal into pragmatic guidance, outlining patient selection criteria for romiplostim/avatrombopag and summarizing ongoing phase III trials that are expected to define registration quality end points [38]. Finally, Song et al. re-emphasized the clinical distinction between transient CIT and persistent CIT, underscoring that the latter drives most bleeding episodes and cycle modifications and is the logical target for TPO-RA-based support [39].

In a recent meta-analysis on metastatic colorectal cancer by Zhan et al. (2024) [23], the Surface Under the Cumulative Ranking (SUCRA) values were used to assess the risk of CIT across different regimens. The analysis found that CAPIRI + bevacizumab (capecitabine and irinotecan with bevacizumab), FOLFIRI + bevacizumab (5-fluorouracil, leucovorin, and irinotecan with bevacizumab), and CAPIRI + cetuximab (capecitabine and irinotecan with cetuximab) had the lowest risk of thrombocytopenia, ranking highest in safety. Conversely, S-1 + oxaliplatin, FUOX, FOLFOXIRI + bevacizumab, and IROX exhibited a higher risk of thrombocytopenia, as reflected by their lower SUCRA scores. Notably, the study also observed that platinum-containing regimens were more strongly associated with thrombocytopenia, highlighting the need for close platelet monitoring in patients receiving these treatments [23]. Among combination regimens, FOLFIRINOX has been linked to a higher incidence of Grade 3–4 thrombocytopenia compared to gemcitabine alone [2] (Table 6).

## 5. Management

Table 7 and Figure 2 frame a stepwise strategy for GI cancer CITs. It uses the World Health Organization (WHO) bleeding scale on the CTCAE platelet threshold, because clinically significant hemorrhage (WHO grade ≥ 2) warrants a higher transfusion goal regardless of the platelet count [40,41]. When platelets are ≥50 × 10^9^/L (CTCAE grade 1–2) and bleeding is absent, prospective studies reveal that continuing chemotherapy with weekly platelet count monitoring is considered safe; both NICE and AABB guidelines recommend no transfusion at these levels [42,43]. Once platelet levels fall to 25–49 × 10^9^/L (grade 3) and the next cycle is more than two weeks away, a short delay or 25% dose reduction is required. When the cycle is imminent, pharmacological intervention is preferred, such as in the pivotal randomized trial of romiplostim, 93% of persistent CIT patients reached ≥100 × 10^9^/L within three weeks versus 12.5% under observation with divergence evident after one week [44]. For oral avatrombopag, a 74-patient multicenter study reported a median 9.4 days to achieve ≥75 × 10^9^/L and a platelet transfusion requirement of only 18.9% despite all participants starting at ≤50 × 10^9^/L [45]. When platelets are <25 × 10^9^/L (CTCAE grade 4), this requires immediate halting of chemotherapy and platelet transfusion when platelets drop below 10 × 10^9^/L or if there is bleeding, and TPO-RA is started on day 1 to shorten time to recovery. If concurrent WHO ≥2 bleeding is present, both NICE and AABB guidelines recommend transfusing to ≥50 × 10^9^/L, after which TPO-RA therapy and reassessment every 48–72 h are advised [42,43].

Figure 2 presents an evidence-based workflow for GI cancer CIT management. The algorithm first stratifies patients by WHO bleeding grade because clinically significant hemorrhage supersedes platelet count thresholds. In the absence of WHO ≥ 2 bleeding, CTCAE platelet grades dictate action: platelet counts ≥50 × 10^9^ permit full dose chemotherapy with monitoring, grade 3 counts trigger dose modification or a five-day course of avatrombopag when the next cycle is imminent, grade 4 counts mandate transfusion if <10 × 10^9^ and immediate TPO-RA initiation. All pathways converge on weekly reassessment, ensuring timely re-escalation of chemotherapy once platelets recover to ≥50 × 10^9^. This structure mirrors thresholds used in the RECITE and avatrombopag phase III trials while aligning with ISTH transfusion guidance [46].

Previous reviews have provided broad overviews of CIT, covering epidemiology, mechanisms, and general TPO-RA guidance, but none of these reviews have focused their scope on gastrointestinal cancers specifically [6,38]. Ju Li et al. developed and validated a nomogram in 750 GI-malignancy patients receiving oxaliplatin-based therapy, identifying metastatic stage, total oxaliplatin dose, albumin, baseline platelet count, and natural killer cells percentage as independent CIT predictors (area under ROC curve (AUC) 0.877) [47]. In another multicenter retrospective series studying 58 GI cancer patients treated with avatrombopag, it was evident that the recovery rates were significant given the platelet recovery rates of 53.4%, 82.8%, and 96.6% by days 5 ± 2, 11 ± 3, and 18 ± 3, respectively, with no significant drug-related toxicity [36]. Most importantly, the phase III RECITE trial (NCT03362177) of romiplostim versus placebo in oxaliplatin-treated GI cancers demonstrated that 84% of romiplostim-treated patients avoid CIT-induced chemotherapy dose modifications when compared with the 36% in the placebo group (OR 10.2; 95% CI 4.6–22.5; *p* < 0.001) [48].

Beyond these, eltrombopag was evaluated in a randomized placebo-controlled phase II study (NCT04600960) where it proved to reduce grade 3/4 thrombocytopenia rates in combination therapy (77% vs. 100%) and monotherapy (36% vs. 42%), shorten mean time to platelet nadir recovery from 15 to 8 days, and lower chemotherapy dose delays/reductions in both combination (77% vs. 91%) and monotherapy (62% vs. 83%) [49]. Likewise, hetrombopag in a multicenter double blind phase II trial (NCT03976882) showed a treatment response rate of 60.7% versus 12.9% with placebo (OR 10.44; 95% CI 2.82–38.65; *p* < 0.001), with comparable grade ≥ 3 adverse event rates (17.9–35.7% vs. 19.4–35.5%) [50].

The management of CIT requires a comprehensive approach, considering the underlying cause, chemotherapy regimen, and treatment goals. Before initiating specific interventions, it is crucial to evaluate for secondary causes of thrombocytopenia, such as infection, coagulopathy, bone marrow suppression, or concurrent medications that may exacerbate platelet suppression [18]. In CIT, depending on chemotherapy regimen and risk of myelosuppression, platelet nadir and recovery varies. However, the depth of the platelet nadir may worsen with successive chemotherapy cycles, leading to an increased risk of bleeding and the need for treatment modifications [31].

One major strategy is to reduce the chemotherapy frequency or dosage. This is usually preferred if the therapy is not standard or not of curative intent [9]. A retrospective analysis evaluating irinotecan-based (FOLFIRI) and oxaliplatin-based (mFOLFOX6) regimens in metastatic colorectal cancer patients demonstrated that maintaining higher relative dose intensity (RDI) of irinotecan was significantly associated with improved outcomes (PFS: 9.9 vs. 5.6 months and OS: 26.7 vs. 12.9 months) [51]. These findings underscore the critical balance between managing CIT and preserving chemotherapy dose intensity to optimize patient outcomes.

Platelet transfusion is a key intervention in the management of CIT, particularly for patients at high risk of bleeding or those experiencing active hemorrhage. According to the American Society of Clinical Oncology (ASCO) guidelines, prophylactic platelet transfusions are recommended when platelet counts fall below 10 × 10^9^/L in solid tumors. A higher threshold can be used in case of bleeding or necrotic tumors [52]. A few studies mention <20 × 10^9^/L if the patient is febrile [18]. However, platelet transfusions have several limitations, making their use less ideal for long-term management. One of the main concerns is their short-lived effect—transfused platelets survive only 3–5 days [9,53]. Additionally, repeated transfusions can lead to alloimmunization, where patients develop HLA antibodies, making subsequent transfusions less effective and increasing the risk of platelet refractoriness. Patients who become refractory may require HLA-matched platelets [53]. Furthermore, platelet transfusions carry risks of transfusion-related reactions, infections, and thrombotic complications [54]. Given these limitations, platelet transfusions should be used judiciously, mainly for acute management of severe thrombocytopenia or active bleeding.

In patients with liver cirrhosis, splenic sequestration often contributes to persistent thrombocytopenia, which can complicate the administration of systemic chemotherapy. Partial splenic embolization (PSE) offers a minimally invasive strategy to mitigate hypersplenism by reducing splenic blood flow, thereby increasing circulating platelet counts. This approach has shown effectiveness in improving hematologic parameters, enabling safer delivery of chemotherapy in select cirrhotic patients. A prospective phase II study demonstrated that PSE enabled 94% of patients with gastrointestinal cancers to resume chemotherapy within a median of 14 days post-procedure, significantly improving platelet counts and minimal procedure-related morbidity [55]. Compared to splenectomy, PSE carries a lower procedural risk and can be tailored to minimize complications such as infarction or portal vein thrombosis [56].

Antifibrinolytic agents like ε-aminocaproic acid and tranexamic acid have been considered for managing bleeding in thrombocytopenic cancer patients when platelet transfusions are ineffective. However, their clinical benefit remains unproven, and their use may increase thrombotic risk, particularly in cancer patients [9,18].

Thrombopoietin receptor agonists (TPO-RAs) have emerged as a promising therapeutic option for managing CIT, particularly in patients with solid tumors. These agents, including romiplostim, eltrombopag, and avatrombopag, function by stimulating the thrombopoietin receptor, thereby enhancing megakaryocyte proliferation and increasing platelet production. Their use aims to maintain chemotherapy dose intensity, reduce the need for platelet transfusions, and mitigate bleeding risks associated with CIT.

TPO-RAs are a class of drugs that bind to and activate the TPO receptor (MPL), stimulating megakaryocyte proliferation, differentiation, and platelet production, without containing the peptide sequence of endogenous thrombopoietin [57]. There are currently four available TPO-RAs: romiplostim, a “peptibody” administered via weekly subcutaneous injection, and eltrombopag, avatrombopag, and lusutrombopag, which are oral small-molecule agents [46]. While TPO-RAs are FDA-approved for conditions such as immune thrombocytopenia (ITP), hepatitis C-associated thrombocytopenia, aplastic anemia, and periprocedural thrombocytopenia in chronic liver disease, their role in CIT remains under investigation. To date, only studies involving romiplostim have shown a significant benefit in CIT, whereas other agents require further evaluation [46].

Romiplostim, a subcutaneous TPO-RA, has demonstrated CIT, enabling chemotherapy continuation and reducing the need for platelet transfusions (Figure 3). In a retrospective study of 20 cancer patients with platelet counts <100 × 10^9^/L and prior chemotherapy dose delays or reductions, romiplostim increased platelet counts in all patients, with 19/20 achieving ≥100 × 10^9^/L and 15 resuming chemotherapy, of whom 14 completed at least two more cycles without modifications [58].

A phase II trial evaluated weekly romiplostim in CIT, where over half of the patients had primary gastrointestinal malignancies, and nearly 50% had primary or metastatic liver involvement. Romiplostim led to platelet recovery (≥100,000/μL) within 3 weeks in 93% (14/15) of treated patients, compared to only 12.5% (1/8) in the observation group. Mean platelet counts increased from 63,000/μL to 141,000/μL, allowing for safe chemotherapy resumption, and none of the romiplostim-treated patients required platelet transfusions. Due to the strong statistical significance (*p* < 0.001) and lack of spontaneous platelet recovery in the control group, the trial was converted into a single-arm, open-label study, where 85% (44/52) of patients achieved platelet correction within 3 weeks. Among those who resumed chemotherapy with romiplostim maintenance, 64% continued the same chemotherapy regimen, and 58% of those who previously required dose reductions were able to return to full or increased dosing. Only 6.8% (3/44) experienced CIT recurrence, leading to dose modifications. 10.2% of patients developed venous thromboembolism (VTE), but romiplostim was not discontinued due to VTE, and there was no observed increase in myocardial infarction or stroke risk. These findings reinforce romiplostim’s potential to restore platelet counts, sustain chemotherapy dose intensity, and minimize transfusion dependency while maintaining a stable safety profile [44].

In addition to romiplostim, several other TPO-RAs, including eltrombopag, avatrombopag, and lusutrombopag, have been explored for CIT. In a phase II study of solid tumors receiving gemcitabine monotherapy or gemcitabine with cisplatin/carboplatin, eltrombopag failed to show a significant improvement in platelet nadir compared to placebo [49]. Similarly, a randomized, double-blind, placebo-controlled, phase 3 study evaluating the use of avatrombopag in non-hematological malignancies showed no difference in CIT between avatrombopag and placebo group [59].

The wide integration of immune checkpoint inhibitors (ICI) with oxaliplatin or irinotecan-based backbones has created a platelet toxicity profile that is mechanistically distinct from classic bone marrow suppression yet remains unprobed. The incidence of immune-related thrombocytopenia (irTCP) ranges from 0.2–2.8%, thus causing platelet counts to decrease exponentially and is often unresponsive to brief dose interruptions, and as of yet no prospective trial has incorporated TPO-receptor agonists or steroids into an ICI-specific management algorithm [60]. From a health economics perspective, each cycle complicated by CIT adds a mean USD 2179 in direct costs, with hospital episodes averaging USD 36,448 for a 4.6 day stay, according to a 215,508-patient U.S. claims analysis [61]. A complementary cohort showed that each cycle complicated by thrombocytopenia cost an incremental USD 1037 rising to USD 3519 (28% of cases) which were very high-cost cycles, largely due to intensive platelet transfusion requirements [62]. These findings underscore the dual clinical and economic rationale for incorporating TPO-RAs into future immunochemotherapy protocols.

## 6. Conclusions

CIT remains a significant challenge in the treatment of gastrointestinal cancers, often leading to treatment delays, dose modifications, and increased bleeding risks. The complex pathophysiology of CIT involves direct bone marrow suppression, thrombopoietin dysregulation, immune-mediated destruction, and chemotherapy-induced microenvironmental changes. The incidence and severity of CIT vary among different chemotherapy regimens, with platinum-based and fluoropyrimidine-containing treatments showing a particularly high risk. Current management strategies include chemotherapy dose adjustments, platelet transfusions, and emerging TPO-RAs, which have shown promise in mitigating platelet depletion while preserving treatment efficacy. However, limitations such as transfusion-related complications and potential resistance to TPO-RAs necessitate further exploration of novel therapeutic options. A multidisciplinary approach incorporating early detection, individualized treatment strategies, and ongoing clinical research is essential to optimizing outcomes for patients with GI malignancies. Future studies should focus on refining risk stratification models and evaluating innovative agents that enhance platelet production while minimizing adverse effects, ultimately improving the quality of life and prognosis for affected patients.

## Figures and Tables

**Figure 1 curroncol-32-00455-f001:**
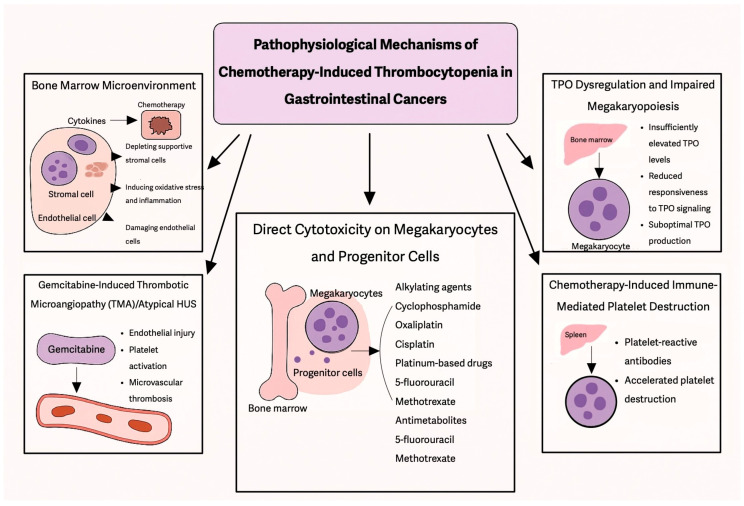
Illustrates the complex and multifactorial pathophysiological mechanisms underlying CIT in GI cancers. Key processes include disruption of the bone marrow microenvironment, where chemotherapeutic agents damage stromal and endothelial cells, alter cytokine signaling, and induce oxidative stress, ultimately impairing hematopoietic stem cell function and platelet production. Direct cytotoxicity to megakaryocytes and progenitor cells by agents such as alkylating compounds, platinum-based therapies, and antimetabolites further compromises thrombopoiesis. In addition, dysregulation of thrombopoietin (TPO) signaling—due to elevated levels with inadequate marrow responsiveness or suboptimal production—impairs megakaryopoiesis. Some patients also experience immune-mediated platelet destruction, where chemotherapy triggers the formation of platelet-reactive antibodies, accelerating platelet clearance via the spleen. Lastly, gemcitabine can induce thrombotic microangiopathy (TMA), characterized by endothelial injury and microvascular thrombosis, contributing to significant platelet consumption. These interconnected mechanisms underscore the clinical complexity of CIT in GI oncology.

**Figure 2 curroncol-32-00455-f002:**
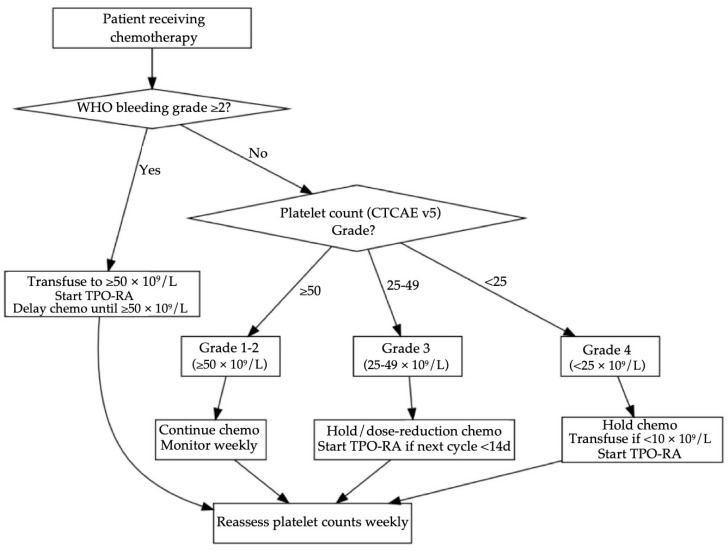
Algorithm for managing chemotherapy-induced thrombocytopenia (CIT) in gastrointestinal (GI) cancers.

**Figure 3 curroncol-32-00455-f003:**
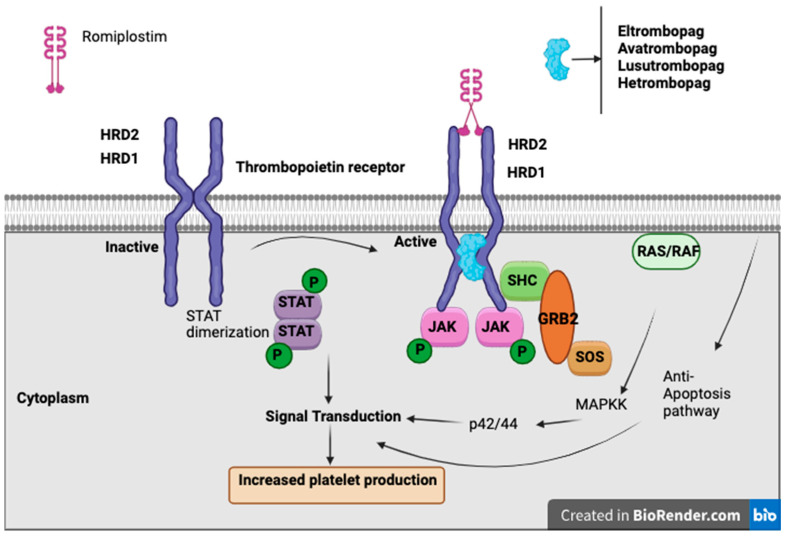
The figure illustrates the molecular mechanism of thrombopoietin receptor activation and its role in promoting platelet production. TPO binds to its receptor, which consists of the HRD1 and HRD2 domains, leading to receptor dimerization and activation. Upon activation, the receptor recruits and phosphorylates JAK kinases, which phosphorylates downstream STAT proteins, leading to their dimerization and subsequent signal transduction for increased platelet production. Additionally, activation of the MAPK pathway via SHC, GRB2, and SOS triggers the RAS/RAF signaling cascade, promoting anti-apoptotic effects. The figure also highlights the pharmacological agents Romiplostim, Eltrombopag, Avatrombopag, Lusutrombopag, and Hetrombopag, which mimic TPO function by binding to and activating the thrombopoietin receptor, thereby enhancing platelet production. These therapeutic agents are clinically used to manage thrombocytopenia by stimulating megakaryopoiesis through JAK-STAT and MAPK signaling pathways. This figure was created with the assistance of BioRender.com.

**Table 1 curroncol-32-00455-t001:** Pathophysiology of chemotherapy-induced thrombocytopenia.

Mechanism	Description	Chemotherapeutic Agents Involved
**Direct Cytotoxicity on Megakaryocytes and Progenitor Cells**	Chemotherapy damages actively proliferating hematopoietic stem and progenitor cells (HSPCs), leading to apoptosis and DNA damage in megakaryocyte precursors.	Alkylating agents (e.g., cyclophosphamide [12]), platinum-based drugs (e.g., oxaliplatin [13]), antimetabolites (e.g., 5-fluorouracil [14])
**Disruption of the Bone Marrow Microenvironment**	Chemotherapy depletes stromal cells, induces oxidative stress, and damages endothelial cells, leading to impaired hematopoiesis and suppressed megakaryopoiesis.	Platinum-based chemotherapy (e.g., cisplatin, oxaliplatin [13]), targeted therapies (e.g., bevacizumab [15], olaparib [16])
**TPO Dysregulation and Impaired Megakaryopoiesis**	Chemotherapy interferes with TPO production and megakaryocyte responsiveness, delaying platelet recovery. Liver dysfunction due to chemotherapy may further reduce TPO synthesis.	Chemotherapy agents, especially those affecting the liver [17] (e.g., high-dose alkylating agents, antimetabolites [18])
**Chemotherapy-Induced Immune-Mediated Platelet Destruction**	Certain chemotherapies and immunotherapies trigger immune responses against platelets, causing their premature destruction via platelet-reactive antibodies or ITP-like mechanisms.	Chemotherapy (e.g., oxaliplatin [19], irinotecan [20]), immune checkpoint inhibitors (e.g., ipilimumab, nivolumab [21]), and targeted therapies (e.g., Trastuzumab [22])

**Table 2 curroncol-32-00455-t002:** Effects of bone marrow microenvironment disruption on CIT.

Affected Component	Impact on Hematopoiesis	Consequence for CIT	References
**Stromal Cells**	Reduced secretion of essential growth factors like TPO and interleukins	Impaired megakaryocyte differentiation and platelet production	Hoggatt et al., 2016 [10]
**Endothelial Cells**	Increased oxidative stress and inflammation	Disrupted bone marrow protection for stem cells, leading to prolonged thrombocytopenia	BMC Cancer, 2024 [23]
**Cytokine Regulation**	Chemotherapy-induced alterations in cytokine balance	Inefficient platelet regeneration post-chemotherapy	Hoggatt et al., 2016 [10]

**Table 3 curroncol-32-00455-t003:** TPO dysregulation and its effects on CIT.

Dysregulation Mechanism	Effect on Platelet Production	Chemotherapy Agents Implicated	References
**Inadequate TPO elevation**	TPO levels do not sufficiently increase to compensate for platelet loss	Alkylating agents, some antimetabolites	Kaushansky, 2016 [24]
**Reduced Megakaryocyte Sensitivity to TPO**	Megakaryocyte precursors fail to respond effectively to TPO stimulation	Platinum-based chemotherapy, targeted therapies	ASCO, 2023 [25]
**Liver Dysfunction Affecting TPO Production**	Impaired hepatic synthesis of TPO leads to prolonged thrombocytopenia	Chemotherapy drugs affecting liver function (e.g., methotrexate, high-dose cyclophosphamide)	Kaushansky, 2016 [24]

**Table 4 curroncol-32-00455-t004:** Immune-mediated platelet destruction.

Mechanism	Associated Chemotherapeutic Agents	Clinical Consequences	References
**Immune checkpoint inhibitor-induced thrombocytopenia**	Pembrolizumab, nivolumab	Increased platelet destruction, ITP-like syndrome	ASH, 2018 [26]
**Drug-associated platelet-reactive antibodies**	Monoclonal antibodies, targeted therapies	Accelerated platelet clearance in spleen, worsening thrombocytopenia	ASH, 2018 [26]
**T-cell-mediated platelet destruction**	Some targeted therapies and immune-based treatments	Increased risk of severe bleeding complications	ASH, 2018 [26]

**Table 5 curroncol-32-00455-t005:** Chemotherapy-induced thrombocytopenia (CIT) and its clinical manifestations [33].

CIT Grade	Platelet Count (×10^9^/L)	Clinical Manifestations	Recommended Intervention
**Grade 1–2**	50–150	Often asymptomatic or subtle signs such as petechiae, minimal bleeding	Mild thrombocytopenia, no intervention required.
**Grade 3**	25–50	Noticeable symptoms, including easy bruising, ecchymoses, and mucosal bleeding	Hold or reduce chemotherapy; consider platelet transfusion if bleeding occurs.
**Grade 4**	<25	High risk of major bleeding (GI tract, intracranial, retroperitoneal hemorrhages)	Immediate platelet transfusion; discontinue causative therapy if needed.

**Table 6 curroncol-32-00455-t006:** Incidence of chemotherapy-induced thrombocytopenia (CIT) in gastrointestinal cancers.

Tumor Type	Chemotherapy Regimen	Incidence of CIT (%)	Severity and Risk Factors	Reference
**Colorectal Cancer (CRC)**	CAPIRI + bevacizumab (capecitabine, irinotecan, bevacizumab)	Low risk (SUCRA ranking highest in safety)	Lower thrombocytopenia risk compared to platinum-based regimens	Zhan et al., 2024 [23]
	FOLFIRI + bevacizumab (5-FU, leucovorin, irinotecan, bevacizumab)	Low risk	Minimal platelet suppression	Zhan et al., 2024 [23]
	CAPIRI + cetuximab (capecitabine, irinotecan, cetuximab)	Low risk	Similar thrombocytopenia risk to FOLFIRI	Zhan et al., 2024 [23]
	S-1 + oxaliplatin (SOX)	High risk	Strong association with thrombocytopenia	Zhan et al., 2024 [23]
	FOLFOXIRI + bevacizumab (5-FU, leucovorin, oxaliplatin, irinotecan, bevacizumab)	High risk	Increased myelosuppression and platelet reduction	Zhan et al., 2024 [23]
	IROX (irinotecan + oxaliplatin)	High risk	Increased risk due to dual cytotoxic effects	Zhan et al., 2024 [23]
**Pancreatic Cancer**	Gemcitabine-based regimens	20–30%	High thrombocytopenia incidence, worsened by combination therapies	Wu et al., 2024 [7]
	FOLFIRINOX (5-FU, leucovorin, oxaliplatin, irinotecan)	20–25% (Grade 3–4 CIT)	Higher thrombocytopenia risk vs. gemcitabine alone	Shaw et al., 2024 [32]
**Gastric Cancer**	FUOX (5-FU, oxaliplatin)	Moderate to High	Platinum-based agents contribute to thrombocytopenia	Zhan et al., 2024 [23]
**Hepatocellular Carcinoma (HCC)**	Platinum-based chemotherapy (cisplatin, oxaliplatin)	30–40%	Liver dysfunction exacerbates platelet depletion	Shaw et al., 2024 [32]
**Esophageal Cancer**	5-FU + Cisplatin	13–25%	Increased thrombocytopenia risk with combination chemotherapy	Wu et al., 2024 [7]

**Table 7 curroncol-32-00455-t007:** Grade and bleeding-directed management of CIT in gastrointestinal cancers.

CTCAE Platelet Grade	WHO Bleeding Grade	Action on Chemotherapy	Platelet Transfusion Threshold	TPO-RA
1 (≥75 × 10^9^/L)	0–1	Continue full dose	None	None
2 (50–74 × 10^9^/L)	0–1	Continue full dose while monitoring	None	Consider if downward trend and next cycle <7 days
3 (25–49 × 10^9^/L)	0–1	Hold or 25% dose	<10 × 10^9^/L	Start avatrombopag (60 mg OD for 5 days) or romiplostim (3–5 µg kg weekly)
3 (25–49 × 10^9^/L)	≥2	Hold chemotherapy	Transfuse to ≥50 × 10^9^/L	Start after bleeding is controlled
4 (<25 × 10^9^/L)	0–1	Hold chemotherapy	<10 × 10^9^/L	Start immediately
4 (<25 × 10^9^/L)	≥2	Hold chemotherapy	Transfuse to ≥50 × 10^9^/L	Start immediately

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
