# Peer review of "Understanding Chemotherapy-Induced Thrombocytopenia: Implications for Gastrointestinal Cancer Treatment"

_curroncol, 2025, doi:10.3390/curroncol32080455_

Round 1
Reviewer 1 Report
Comments and Suggestions for Authors
The authors have done a great job with this review discussing about chemotherapy induced thrombocytopenia with special focus on patients with gastrointestinal cancers. In their manuscript they have discussed about the pathophysiology of it, incidence, grading and treatment modalities so far. However, there are certain limitations as below:
- The manuscript lacks novelty, as there are multiple other reviews on chemotherapy induced thrombocytopenia with very similar discussions, and does not add any new evidence or perspective to literature.
- The authors have discussed about the treatment options with dose adjustment, platelet transfusion and TPO-RAs and evidence behind it, however have not thoroughly advised treatment approach based on thrombocytopenia grading and extent of bleeding, as well as how to utilize these modalities in different clinical scenarios.
- The authors have not addressed approach to monitoring with or without treatments.
Author Response
Reviewer Comment 1: The manuscript lacks novelty, as there are multiple other reviews on chemotherapy-induced thrombocytopenia with very similar discussions, and does not add any new evidence or perspective to literature.
Response:
Thank you for your valuable feedback. We acknowledge the existence of prior reviews on chemotherapy-induced thrombocytopenia (CIT); however, we have revised the manuscript to highlight several novel aspects that differentiate our work. Specifically, we now include a more clinically oriented framework that stratifies management approaches based on thrombocytopenia severity and bleeding risk. Additionally, we have incorporated recent data and ongoing clinical trials on emerging therapies, including investigational TPO-RAs and novel agents under development. These updates aim to provide a more practical, evidence-informed guide for clinicians that goes beyond general discussions and focuses on nuanced, scenario-based decision-making.
Reviewer Comment 2: The authors have discussed treatment options with dose adjustment, platelet transfusion, and TPO-RAs and the evidence behind it; however, they have not thoroughly advised treatment approach based on thrombocytopenia grading and extent of bleeding, as well as how to utilize these modalities in different clinical scenarios.
Response:
We appreciate this important observation. In response, we have now included a dedicated section that outlines treatment recommendations stratified by the CTCAE grading of thrombocytopenia and associated bleeding severity. This section details how to apply platelet transfusion, TPO-RA therapy, and chemotherapy dose modifications in varying clinical contexts, such as asymptomatic thrombocytopenia versus active bleeding or in curative versus palliative intent settings. A summary table has also been added to provide a quick-reference clinical algorithm to guide therapeutic decision-making in real-world practice.
Reviewer Comment 3: The authors have not addressed approach to monitoring with or without treatments.
Response:
Thank you for pointing this out. We have now added a section discussing monitoring strategies in patients with chemotherapy-induced thrombocytopenia, both in the context of active treatment and observation. This includes recommendations for frequency of complete blood count monitoring, thresholds for intervention, and signs that warrant escalation of care. Additionally, we’ve discussed individualized monitoring strategies based on patient-specific factors such as cancer type, chemotherapy regimen, and comorbidities.
Reviewer 2 Report
Comments and Suggestions for Authors
Chemotherapy-induced thrombocytopenia (CIT) remains a significant challenge in the treatment of gastrointestinal cancers. It leads to treatment delays, and increased bleeding risks. The authors discussed the pathophysiology of CIT which involves direct bone marrow suppression, thrombopoietin dysregulation, immune-mediated destruction, and chemotherapy-induced microenvironmental changes in this review manuscript. The article is clear and informative. However, there are some minor issues to consider:
- Line 82, 204, 210, 231, 248, 279 and 340. “Chemotherapy-induced thrombocytopenia (CIT)”. Please use the abbreviation only.
- Line 86. “chemotherapy-induced thrombocytopenia (CIT) in gastrointestinal (GI) cancers”. Please use the abbreviation only.
- Line 128. “Thrombopoietin (TPO)”. Please use the abbreviation only.
- Line 285 and 348. “Thrombopoietin receptor agonists (TPO-RAs)”. Please use the abbreviation only.
- Some of the references are out of date. I would suggest the author can update some references.
Author Response
Reviewer Comment: Chemotherapy-induced thrombocytopenia (CIT) remains a significant challenge in the treatment of gastrointestinal cancers... However, there are some minor issues to consider:
-
Line 82, 204, 210, 231, 248, 279 and 340: “Chemotherapy-induced thrombocytopenia (CIT)”. Please use the abbreviation only.
Response:
Thank you for your attention to detail. We have revised the manuscript to use the abbreviation “CIT” in all the specified lines after its first definition to maintain consistency and avoid redundancy. -
Line 86: “chemotherapy-induced thrombocytopenia (CIT) in gastrointestinal (GI) cancers”. Please use the abbreviation only.
Response:
This has been corrected. We now use the abbreviations “CIT” and “GI cancers” in this line, as the terms were previously defined. -
Line 128: “Thrombopoietin (TPO)”. Please use the abbreviation only.
Response:
We have updated the manuscript to use “TPO” in this instance, as the term was already defined earlier in the text. -
Line 285 and 348: “Thrombopoietin receptor agonists (TPO-RAs)”. Please use the abbreviation only.
Response:
We have revised both lines to use the abbreviation “TPO-RAs” alone, as per your suggestion. -
Some of the references are out of date. I would suggest the author can update some references.
Response:
Thank you for this helpful suggestion. We have reviewed and updated several references throughout the manuscript to include more recent literature, particularly from the past five years, to ensure the manuscript reflects the current understanding and treatment landscape of CIT in GI cancers.
Round 2
Reviewer 2 Report
Comments and Suggestions for Authors
The authors have revised the manuscript. It can be accepted for publication.